# Induction of Periodontitis Using Bacterial Strains Isolated from the Human Oral Microbiome in an Experimental Rat Model

**DOI:** 10.3390/biomedicines11082098

**Published:** 2023-07-25

**Authors:** Diana Larisa Ancuta, Diana Mihaela Alexandru, Maria Crivineanu, Cristin Coman

**Affiliations:** 1Faculty of Veterinary Medicine, University of Agronomic Sciences and Veterinary Medicine, 050097 Bucharest, Romania; albu.dm@gmail.com (D.M.A.); maria_crivineanu@yahoo.com (M.C.); comancristin@yahoo.com (C.C.); 2Cantacuzino National Medical Military Institute for Research and Development, 050096 Bucharest, Romania; 3Center of Excellence in Translational Medicine, Fundeni Clinical Institute, 022328 Bucharest, Romania

**Keywords:** periodontitis, rat, ligature, *Aggregatibacter actinomycetemcomitans*, *Fusobacterium nucleatum*, *Streptococcus oralis*

## Abstract

Periodontal disease is that condition resulting in the destruction of periodontal tissues, bone resorption, and tooth loss, the etiology of which is linked to immunological and microbiological factors. The aim of this study was to evaluate the potential trigger of periodontal disease in a rat model using bacterial species incriminated in the pathology of human periodontitis and to establish their optimal concentrations capable of reproducing the disease, with the idea of subsequently developing innovative treatments for the condition. In this study, we included 15 male Wistar rats, aged 20 weeks, which we divided into three groups. In each group, we applied ligatures with gingival retraction wire on the maxillary incisors. The ligature and the gingival sac were contaminated by oral gavage with a mixture of fresh cultures of *Aggregatibacter actinomycetemcomitans* (A.a), *Fusobacterium nucleatum* (F.n) and *Streptococcus oralis* (S.o) in concentrations of 10^8^, 10^9^, and 10^10^ CFU/mL each for 5 days a week for 4 weeks. During the clinical monitoring period of 28 days, overlapped with the period of oral contamination, we followed the expression of clinical signs specific to periodontitis. We also monitored the evolution of body weight and took weekly samples from the oral cavity for the microbiological identification of the tested bacteria and blood samples for hematological examination. At the end of the study, the animals were euthanized, and the ligated incisors were taken for histopathological analysis. The characteristic symptomatology of periodontal disease was expressed from the first week of the study and was maintained until the end, and we were able to identify the bacteria during each examination. Hematologically, the number of neutrophils decreased dramatically (*p* < 0.0001) in the case of the 10^9^ group, unlike the other groups, as did the number of lymphocytes. Histopathologically, we identified neutrophilic infiltrate in all groups, as well as the presence of coccobacilli, periodontal tissue hyperplasia, and periodontal lysis. In the 10^9^ group, we also observed pulpal tissue with necrotic bone fragments and pyogranulomatous inflammatory reaction. By corroborating the data, we can conclude that for the development of periodontal disease using A.a, F.n, and S.o, a concentration of 10^9^ or 10^10^ CFU/mL is required, which must necessarily contaminate a ligature thread applied to the level of the rat’s dental pack.

## 1. Introduction

Periodontitis, or periodontal disease (PD), is an inflammatory immune condition caused by bacterial biofilms developed in the subgingival space that causes the destruction of the connective tissue, destruction of the alveolar ligament, and, ultimately, bone resorption, resulting in the loss of teeth [1]. Therefore, the persistence of oral micro-organisms at the level of dental structures causes an imbalance in bone metabolism, resulting in the release of proinflammatory mediators, growth mediators, and signaling molecules. Periodontitis is the most common cause of tooth loss in the human population and is associated with atherosclerosis, carotid stenosis, premature birth, and low-birth-weight fetuses if from mothers with periodontitis [2,3]. Subsequently, periodontal bacterial colonization in children increases with age, and the bacterial flora in children is similar to that of their mothers [4].

Among the frequent causes associated with periodontitis, we find smoking, diabetes, stress, age, social status, and genetic factors [5], all of which affect the homeostasis of the oral cavity, ultimately leading to disturbances in the oral microbiome [6]. As a result, inflammation and biofilm formation occur on the tooth surface, as well as invasion of the gingival tissues [2]. In dental plaque, bacteria such as *Streptococcus* spp., which are frequently found in the mouth of all people and express adhesins, provide the necessary support for other bacterial colonizers.

Since the pathogenesis of PD has been studied for a long time, today, data are available to characterize the microbial flora both in healthy and affected patients [7]. *Porphyromonas gingivalis* (P.g), *Tannerella forsythia* (T.f), *Treponema denticola* (T.d), and A.a are the most incriminated pathogens that trigger periodontitis. F.n is also involved in periodontal health [8,9] as demonstrated by its frequent detection in subgingival plaque samples, with an important role in the organization of biofilms as a result of the expression of multiple adhesins [10]. PD symptomatology varies according to the patient’s age; thus, in the case of young people, it is mainly triggered by A.a, whereas in adults, the triggering agents of PD are P.g, T.d, and spirochetes. In an anaerobic environment, as we find in the subgingival space, periodontal pockets are formed, where pathogens and pathobionts that express virulence factors grow, leading to imbalances in the host’s inflammatory response [11].

A.a is a non-motile Gram(−) facultative anaerobe belonging to the *Pasteurellaceae* family [12] that contributes to the occurrence of PD due to several virulence factors that it expresses, such as cytolethal distention toxin, leucotoxin A, and collagenase [13]. In the case of juvenile periodontitis, leukotoxin A is the most studied virulence factor because it kills the polymorphonuclear cells (PMNs)and macrophages, important components of host defense [14], through a proinflammatory process called pyroptosis [15,16]. Under conditions of oral existence of commensals such as F.n and S.o, A.a has a devastating action because in the early stages of periodontitis, A.a uses the lactic acid produced by *Streptococcus* spp. as a nutrient to increase its number [17]. The production of H_2_O_2_ by *Streptococcus* spp. causes A.a to migrate deeper into the gingival pocket, where the bacterial cells are exposed to the host’s immune response [14]. A.a activates the transcriptional regulator of oxygen resistance, which regulates the expression of the Outer membrane protein 100 (Omp100—produced by A.a in response to H_2_O_2_) and catalase [18]. The latter contributes to the degradation of H_2_O_2_ produced by neutrophils and streptococci, protecting A.a from oxidative damage [19]. The release of cytolethal distension toxin in this environment inhibits phagocytosis, and the release of leukotoxin by A.a promotes neutrophil degranulation or death, which leads to the promotion of bone resorption [20].

The purpose of our research was to study the potential trigger of PD in a rat model using the bacterial species incriminated in the pathology of human periodontitis and to establish their optimal concentration capable of reproducing the disease, with the idea of subsequently developing innovative treatments for the condition. To achieve our objectives, we chose S.o, F.n, and A.a as bacterial species. The selection was based on the characteristic of streptococci with respect to its involvement in the early formation of bacterial plaque [21], as well as the fact that F.n is a predominant bacterium that contributes to the formation of biofilm [22] and because A.a is the major pathogenic agent of PD and inflammation [23].

## 2. Materials and Methods

### 2.1. Ethics Statement

The animal experiments were carried out at the Baneasa Animal Facility (BAF) of the Cantacuzino National Medical Military Institute for Research and Development, Bucharest, Romania (CI). The study was approved by the Ethics Committee of the Faculty of Veterinary Medicine, Bucharest, (no. 25/15.06.2022) and by the Romanian competence authority, in accordance with EU Directive 63/2010 on the care, use, and protection of animals used for scientific purposes.

### 2.2. Processing of Bacterial Strains Selected for Study

A.a (ATCC 29522), serogroup b, isolated from a mandibular abscess, was provided by the CI bacterial strain bank. Thus, a cryotube with 1 mL of A.a was revitalized by inoculating a tube with Schadler broth medium, which was incubated for 24 h at 37 °C under anaerobic conditions (95% O_2_ and 5% CO_2_). The density of the 24 h suspension was measured using a densitometer (Densitometer McFarland Biosan DEN-1, Riga, Lithuania), like that of a tube with unseeded medium. The difference determined the concentration of the A.a strain, which corresponded to a concentration of 10^9^ CFU/mL. A concentration of 10^8^ CFU/mL was produced decimal dilution and centrifugation (×4000 rpm, 10 min); with removal of the supernatant, the concentration of 10^10^ CFU/mL was also established. The revitalized A.a culture was stored in cryotubes at −80 °C, which were used to make the daily inoculum throughout oral contamination, that is, for each inoculation, one tube with A.a from the “mother” culture was used.

F.n (ATCC 25586), was isolated from a cervicofacial lesion, and S.o (DSM 20627) was isolated from the mouth of a human patient. A.a came from the CI bacterial strain bank. The inocula were prepared by following the same steps as in the case of A.a, using the same culture media, cultivation conditions, and steps to establish the inoculum density.

For oral contamination, we established 3 concentrations of each strain, namely 10^8^, 10^9^, and 10^10^ CFU/mL, with the inoculum dose established as 0.6 mL (thus, the concentrations per animal were 10^7^ CFU/animal, 10^8^ CFU/animal, and 10^9^ CFU/animal, respectively). The dose at which the 3 bacteria were found in equal volumes (0.2 mL of each bacteria and each concentration) were administered by gavage 5 days/week for 4 weeks.

### 2.3. Periodontitis Rat Model Protocol

The procedures developed to create the animal model for PD were performed on 15 Wistar rats, aged 20 weeks, from the CI Specific Pathogen Free (SPF) animal facility. Throughout the experiment, the animals were housed in groups of 5 under conventional conditions at a temperature of 20–22 °C and under a 12H:12H light–dark cycle and received water and feed ad libitum. Rats were provided with 20 mg/mL kanamycin and 20 mg/mL ampicillin in their drinking water for five days to suppress the resident flora. At the end of the treatment, cotton swabs with saliva were taken from the mouths of the animals to determine the effectiveness of the decontamination and of the flora left unaffected by the treatment. In order to induce periodontitis, we resorted to the application of ligatures on the upper incisors, using a gingival retraction thread (Ultrapak, UltraDent, Bucharest, Romania). Thus, the rats were deeply anesthetized with ketamine (0.5 mg/kg, Pasteur Institute, Bucharest, Romania) and medetomidine (0.5 mg/kg, Biotur, Bucharest, Romania) in a weight-dependent dose. The animals were positioned in ventrodorsal recumbency on the operating table. By applying a mouth spacer, the incisors were isolated. With a dental curette, the gum was detached, and a ligature was applied around each incisor in the bag thus created. The animals were divided into 3 groups depending on the bacterial concentration used (10^8^, 10^9^, and 10^10^ groups). At the end of the procedure, the fresh 24 h inoculum consisting of the three bacteria was used to impregnate the thread and to wash the gingival pocket (Figure 1). Then, the animals received atipamezole (0.02 mg/kg, Biotur, Romania) to reverse the effect of anesthesia.

At the time of each intervention, the body weight and the periodontal pocket were monitored, and blood samples were taken from the retro-orbital sinus to perform a hematological examination. At the end of the study, the animals were euthanized by an overdose of anesthetic, and samples were collected for microbiological examination, as well as the incisors for histopathological analysis.

### 2.4. Statistical Analysis

Analyses were performed using Prism 9 software for Windows (GraphPad LLC, Chicago, IL, USA). To compare the data, the one-way ANOVA function was used, and a value of *p* < 0.05 was considered statistically significant. Regarding the analysis of the data obtained after the hematological examination, we compared the results obtained from each group (10^8^, 10^9^, and 10^10^), comparing them with day 0 using the one-way ANOVA function and multiple comparisons and comparing the data from days 7, 24, and 28 with those from day 0 using Dunnet’s test.

## 3. Results

From a clinical point of view, the animals did not present discomfort during mastication, and through the weekly monitoring of body weight, we observed a relatively upward trend in the first three weeks, with a slight weight decrease recorded in the last week (Figure 2). Gingival bleeding and periodontal pocket formation were clinically visible starting in the second week after contamination.

The hematological examination was performed on an Idexx Procyte 5diff analyzer using blood collected in EDTA vacutainers (KIMA Vacutest, Arzergrande, Italy). We followed the PMNs, since in PD, they are the cells responsible for the annihilation of pathogens. Regarding the number of neutrophils, which represent 50–70% of the total PMN, a decrease in their number was observed as periodontitis was installed in the case of all groups, but a strongly statistically significant relevance was recorded in the 10^9^ group (*p* < 0.0001; Figure 3) in the last 2 weeks of the study. The values expressed on day 0 can be explained by the fact that the blood was collected after finishing the application of ligatures to all animals. Thus, after we finished applying the ligature to the last animal, the blood collection started from animal number 1. Therefore, the time elapsed from the application of the ligatures to the time of blood collection influenced the entry of and the number of neutrophils as the body’s response to the trauma and to the presence of bacteria. The initial response to acute periodontal inflammation is the physiological response to the oral microbial challenge to recruit leukocytes to sites of infection [24]. Regarding the expression of and the number of neutrophils, our analysis focused on the body’s response within the group. In the case of group 10^9^, an increase in the number of neutrophils can be observed from the first blood collection (approximately 3 h after the application of the ligature and contamination with the bacterial inoculum). This is a hyperactive response that occurred as a result of the body’s interaction with micro-organisms, which most likely led to the release of proinflammatory cytokines from the tissues. The same observation was made by Matthews et al. (2007) in a comparative study showing hyperfunctionality of blood PMNs in PD patients compared to healthy controls [25].

Lymphocytes, the elements involved in the immune response of the host, did not provide statistically significant results between day 0 and the final day of the study (*p* < 0.05) except in the case of group 10^9^ (Figure 4), a sign that this group is maintained in an active phase, an aspect strengthened by the activity of neutrophils.

Histological analysis—The samples represented by the maxillary incisors with the ligature, together with the related gum, were collected, fixed in 10% neutral buffered formalin, and demineralized in 5% nitric acid for 14 days. After decalcification, the specimens were dehydrated and embedded in paraffin. Sections with a thickness of 4 μm were obtained in the transverse plane. Sections were stained with hematoxylin and eosin (H&E) using standard protocols [26]. The sections were evaluated by light microscopy (4× magnification), and parameters such as the influx of inflammatory cells and the integrity of the alveolar bone and cement were monitored (Figure 5 and Figure 6).

The microbiological examination consisted of taking samples from the gum and the ligatures. The samples were seeded in a liquid Schadler medium and incubated under anaerobic conditions for 24 h after each renewal of the ligatures. From the 24 h suspension, plates were inoculated with Shadler agar, as well as Columbia 5% ram blood and Columbia 7% ram blood. A 24 h reincubation of the plates followed; then, smears were made from the grown colonies to identify the bacterial strains used. From the first week after contamination, in the case of all groups, Gram(−) specific A.a coccobacilli, F.n Gram(−) bacilli, or S.o Gram(+) chains were identified in the smears, together with the oral microflora of rats represented by *Staphyococcus sciuri*, *Staphylococcus xylosus*, *Proteus mirabilis*, or *Enterococcus faecium*, with the analysis of the smears completed using a MaldiTof (Bruker MALDI Biotyper).

## 4. Discussion

The term periodontitis is used to express the presence and multiplication of micro-organisms at the level of the oral cavity—more precisely, at the level of the gum, ligament, and alveolar bone [27]. Incriminated in the development of PD are Gram(−) anaerobes, the most widespread of which, in the subgingival space, seem to be A.a, P.g, *Prevotella intermedia* (P.i), and T.f. Through an immunopathogenic mechanism, they are involved in the development of the disease from the beginning by multiplying, resulting in the periodontal pocket. The body responds by forming the inflammatory infiltrate represented by macrophages and lymphocytes, which produce cytokines and biological mediators [28].

Dental plaque is a favorable environment for the multiplication of microorganisms and the formation of biofilms [29], so PD is associated with plaque, and although it has a wide etiology, the most studied causes are microbial and immunological [30,31]. In the oral environment, bacteria grow in complex polymicrobial associations, with more than 700 bacterial species living in the oral cavity [32]. Species of the genus *Streptococcus* are early colonizers of the mouth that actively recruit bacteria such as P.g through several genetic mechanisms [33], contributing to the general functional heterogeneity of the biofilm. This heterogeneity provides the biofilm new characteristics, such as easy adhesion to surfaces, metabolic co-operation whereby the waste product of one bacterial species serves as a food source for another [34], increased antibiotic resistance, and the ability of biofilms to evade the host’s immune system. Recent research has shown that A.a, in associated with *Streptocoocus* spp., stimulates resistance to the host’s innate immunity [35], generating imbalances in the normal flora of the oral cavity. This phenomenon is translated by the term dysbiosis, and as periodontitis develops, the oral microbiota changes from one consisting mainly of Gram(+) aerobes to a constant microbiota mainly comprising Gram(−) anaerobes [36], ultimately resulting in the clinical expression of the disease. Simultaneously, a succession of microbial complexes takes place, the first of which is the so-called “orange complex”, which consists of anaerobic Gram(−) species, among which we find F.n [37,38]. F.n is a binder for other bacterial species responsible for PD development, i.e., strains belonging to the “red complex”, which includes bacteria such as P.g, T.f, T.d, and, more recently, A.a [39]. In this sense, the objective of this study was to induce periodontitis through the oral contamination of rats with bacteria that are directly responsible for this condition in humans in order to better understand its pathogenesis and develop therapeutic schemes.

The analysis of periodontitis in a rat model provided remarkable insights into the pathogenesis of the disease, recapitulating the clinical or histological characteristics [40,41]. Rats resemble humans when developing periodontitis in terms of the composition of the dental plaque and the appearance of histopathological lesions specific to this disease. In rats, PD appears within a few weeks when induced by ligatures and in an even shorter period of 7–15 days when the pathogenic bacterial flora intervenes [42].

In most studies, placing a silk thread around the bundle of maxillary or mandibular premolars is reported to stimulate bacterial colonization and biofilm formation, resulting in apical epithelial migration and bone loss as observed in clinical settings [43]. In our study, we placed these ligatures with gingival retraction wire around the maxillary incisors for the advantage of easy access and injury of the gum to apply the ligatures contributed to the rapid establishment of PD, an aspect also mentioned by other researchers who associate traumatic injuries with the pathogenesis of PD induced by ligation in rodents [44]. A simple ligature without bacterial involvement does not cause significant bone loss in rats, as shown by Bezerra et al. [45], who concluded (in contrast to other researchers) that the accumulation of bacteria around the ligature thread plays an important role in PD induction and progression [46]. In light of these considerations with respect to PD induction, we chose the ligation model completed by oral contamination with three of the most representative bacteria for periodontitis. The clinical signs observed after the first week of contamination (bleeding when palpating the gums) suggested the onset of PD installation. A disadvantage of placing the ligature on the maxillary incisors is the loss of the ligatures within 4–5 days of application, even with the ligature in an ”8” pattern [47,48], requiring renewal every week.

Biological mediators involved in periodontitis provide valuable information about host–microbial interactions and inflammation [49]. The diseased periodontal tissue constantly guides neutrophils and leukocytes [50] to the junctional epithelium that borders the oral microflora, causing the activation of immune cells such as lymphocytes. The latter trigger the release of prostaglandins, interleukin-1β (IL-1β), tumor necrosis factor-α (TNF-α), and IL-6, culminating in the triggering of osteoclastogenesis and bone destruction by direct stimulation of osteoclasts or by the release of enzymes for tissue destruction by inflammatory cells [51]. Using the ligature-induced periodontitis model periodically impregnated with A.a, F.n, and S.o in rats, we addressed the host response at clinical, biological, and histological levels from the onset of the disease until the end of the 28-day experimental period, and our results provide evidence of PD installation and the fact that the disease was in full evolutionary process in the group in which the bacterial strains were tested at a concentration of 10^9^.

From a histopathological point of view, the results indicate that the bacterially contaminated ligation pattern leads to progressive alveolar bone resorption, suggesting two distinct phases: an acute phase (in group 10^9^) and a chronic phase (in the case of group 10^10^). Molon et al. divided the process of ligation-induced periodontitis in rats into two successive processes, concluding that in the acute process of periodontitis, inflammatory cell infiltration was obvious and that alveolar bone resorption was rapid, whereas in the chronic phase, the number of infiltrating inflammatory cells decreased, and alveolar bone resorption slowed [47]. In the case of all the tested groups, different stages of periodontitis installation were observed, but the need to differentiate PD installation as a shoulder of ligature placement versus the bacterial action is imperative. Because bone resorption in the ligature model is dependent on the presence of oral micro-organisms [44], in our study, we observed the presence of bacteria in the periodontal sac or attached to the ligature wire, regardless of the concentration used. Clinical periodontitis is mainly an inflammatory disease caused by bacteria as the initiating factor. Through the formation of bacterial plaque, inflammatory cells infiltrate the local periodontal tissue, and the differentiation of osteoclasts occurs, resulting in alveolar bone resorption [52]. Similar to the traditional rat model of periodontitis, this model induced local periodontitis by simulating bacterial aggregation in the periodontal tissue, causing alveolar bone resorption [53]. Bone resorption occurred only in the case of concentrations of 10^9^ and 10^10^, complemented by a pyogranulomatous reaction specific to bacterial aggressiveness on the periodontal tissues, as mentioned by Bascones-Martínez in his research [28]. Comparing the histological effects produced by the three bacterial concentrations, we mention that the severity of the disease depends on the concentration in the sense of exacerbating its intensity, as suggested by Yuan et al. [54]. Therefore, for oral contamination over a period of 4 weeks, the bacterial action is necessary for a concentration of 10^9^ or 10^10^ through the PD model thus created, and advanced bone loss can be expected in a relatively short time.

## 5. Conclusions

In summary, our findings obtained in rats provide experimental evidence that ligations impregnated in bacterial culture represented by A.a, F.n, and S.o induced obvious PD when concentrations of 10^9^ and 10^10^ CFU/mL of each strain were used. Based on the presence of inflammatory infiltrate and bone resorption observed upon histopathological examination, complemented by the abundance of neutrophils in group 10^9^, we recommend this model for the most effective induction of periodontitis in the shortest time.

## Figures and Tables

**Figure 1 biomedicines-11-02098-f001:**
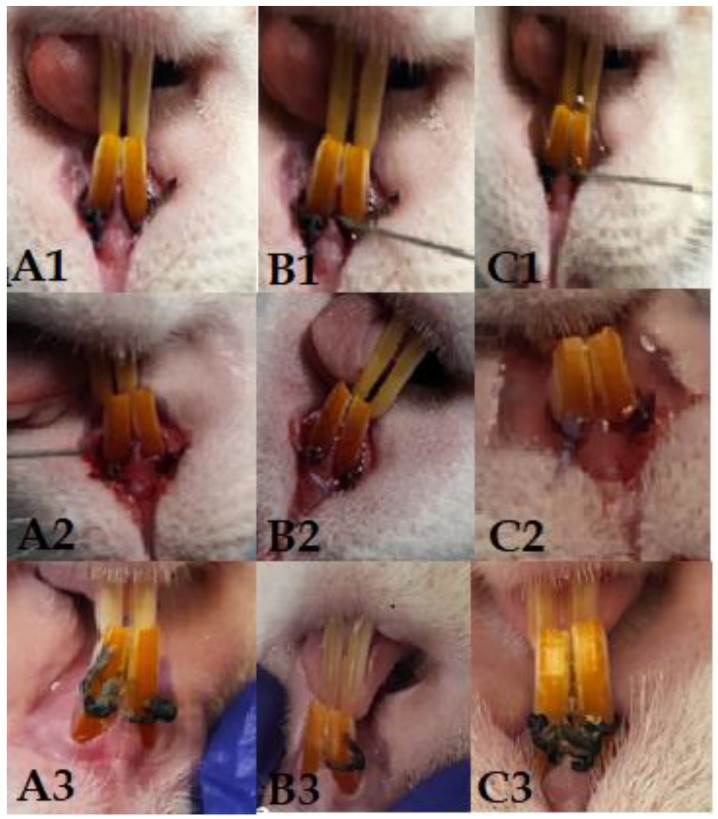
Columns (**A1**–**A3**) represent the 10^8^ group, with clinical aspect at the time of application of the first ligature (**A1**), after 14 days of contamination (**A2**), and on day 28 (**A3**). Columns (**B1**–**B3**) represent the 10^9^ group, with clinical aspect at the time of application of the first ligature (**B1**), after 14 days of contamination (**B2**), and on day 28 (**B3**). Columns (**C1**–**C3**) represent the 10^10^ group, with clinical aspect at the time of application of the first ligature (**C1**), after 14 days of contamination (**C2**), and on day 28 (**C3**).

**Figure 2 biomedicines-11-02098-f002:**
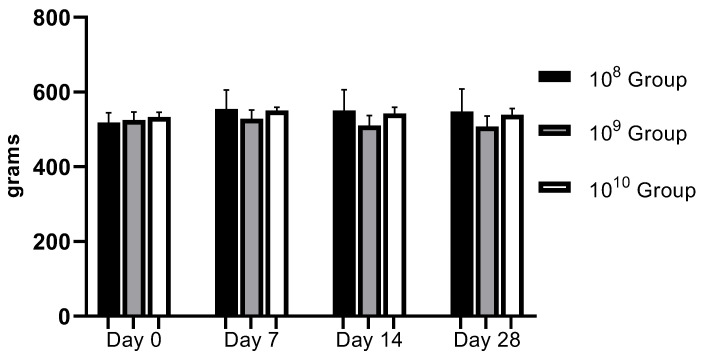
Evolution of body weight during the study (regardless of the bacterial concentration used, no statistically significant changes were recorded).

**Figure 3 biomedicines-11-02098-f003:**
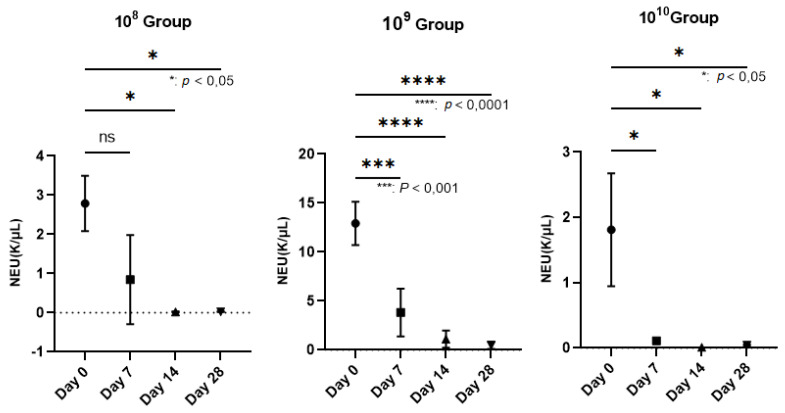
Reduction in the number of neutrophils (NEU) on the wall of the PD installation depending on the bacterial concentration used (In group 10^8^, neutrophils decreased starting the 3rd week after contamination; *p* < 0.05). The most relevant decrease in neutrophils was observed in group 10^9^, where, starting the second week, their number began to decrease more dramatically than in the case of group 10^8^, registering increasingly lower values until the end of the study, when *p* < 0.0001. In group 10^10^, the number of neutrophils decreased constantly during the study, with a *p* value < 0.05.

**Figure 4 biomedicines-11-02098-f004:**
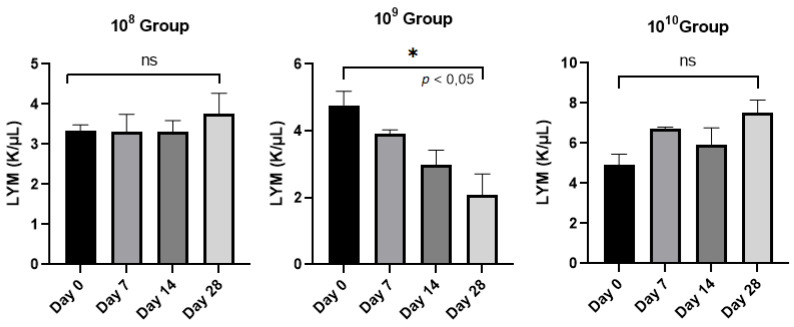
Evolution of the number of lymphocytes (LYM)/group analyzed from day 0 to day 28.

**Figure 5 biomedicines-11-02098-f005:**
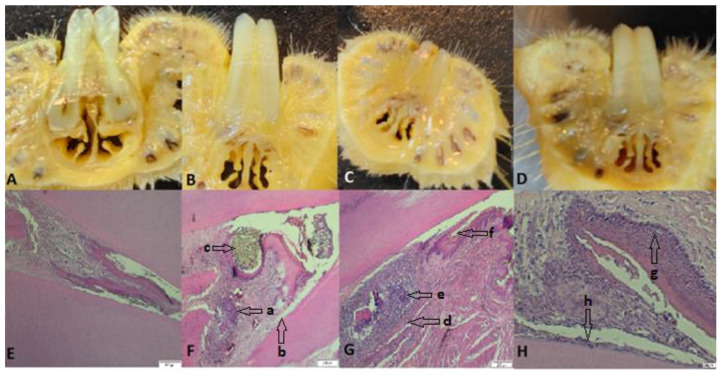
(**A**–**D**) Images with cross sections through the collected part; (**E**) the part collected from a control rat, uncontaminated and without ligature, showing intact alveolar bone and continuous cementum; (**F**) the part from a rat from group 10^8^, which presents hypertrophy, granulation tissue (a) with nucleated cells that peel off, abundant neutrophils, coccobacilli, inflammation, hyperemia, hyperplasia of the gingival epithelium, and periodontal lysis (b), with a ligature thread present (c); (**G**) rat teeth from group 10^9^ (granuloma) (d), alveolar bone lysis, pulp tissue with necrotic bone fragments surrounded by pyogranulomatous inflammation, neutrophilic infiltrate (e), abundant granulation tissue, bacteria (coccobacilli), and fodder in the space periodontally invaded with bacteria (f); (**H**) teeth of rats from group 10^10^ with hyperplasia (g), an intact alveolodental ligament, abundant granulation tissue, fewer inflammatory cells, bacteria in the alveolar sac, and anucleated desquamation cells on the alveolodental ligament area (h). Sections were stained with H&E. Original magnification: 4×; scale bars = 200 μm.

**Figure 6 biomedicines-11-02098-f006:**
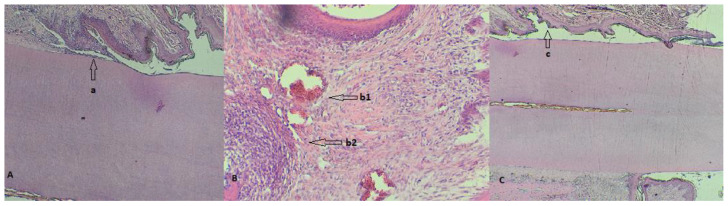
(**A**) Detail of a rat tooth from group 10^8^ with a partially desquamated alveolodental ligament (a); (**B**) strong reaction of hyperemia (b1) and neutrophilic infiltrates (b2) in a rat from group 10^9^; (**C**) overview highlighting the periodontal sac (c) of a rat from group 10^10^.

## Data Availability

Not applicable.

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
