# Peer review of "Induction of Periodontitis Using Bacterial Strains Isolated from the Human Oral Microbiome in an Experimental Rat Model"

_biomedicines, 2023, doi:10.3390/biomedicines11082098_

Round 1

Reviewer 1 Report

Induction of periodontitis using bacterial strains isolated from the human oral microbiome in an experimental rat model

By ancuta et al.

In this paper, the authors demonstrate the development of a model for inducing periodontitis in the shortest and most effective time by oral contamination with three of the most representative bacteria for periodontitis. The paper is on an important topic and is a significant development in the field. I have the following additional comments on the manuscript.

Line 19: “4 days/week, 4 weeks,” Grammatically incorrect

Line 20: Change “with fresh cultures of” to “with a mixture of fresh cultures of” (Although never mentioned, I guess the three strains were mixed in the same inoculum)

Line 21: Change “CFU/ml” to “CFU/ ml each” (Although not clearly mentioned, I guess that each mixture had the three strains, each at the mentioned concentrations)

Line 20: Change “contaminated with” Since the abstract is also presented independent of the rest of the paper, please mention very briefly how the inoculation was done.

Line 22: “monitoring period was 28 days” Is this the same as the 28 days of inoculation or is this after inoculation?

Line 29: “group 109” The 9 should be in superscript

Line 34: “CFU” Space before CFU

Line 46: “is associated with ……. premature birth, low birth weight fetus,” Meaning of this is not clear. Does PD in the mother result in these conditions in the offspring or do these conditions during birth cause PD later when adult?

Line 50-52: “Consequently ….. tissues” Grammatically incorrect.

Line 67 and throughout the manuscript: “Gram-” the negative sign can be mistaken for a hyphen especially when not followed by a space. Possible solutions can be to write “Gram negative” or “Gram (-)”

Line 68: Change “factors it expresses” to “factors it expresses, such as,”

Line 71-74: “In the conditions….. increase its number.” Please provide a reference for this.

Line 74-76: “The production of…..immune response.” Please provide a reference for this. Also, doesn’t Aa produce catalase to counteract the effect of H2O2?

Line 92: “originating from the collection of bacterial strains of IC” Do these strains have any formal identification numbers in the IC catalog (similar to ATCC numbers)? Can researchers from any lab in the world have access to strains from IC? Is the Aa strain used, of the rough phenotype or smooth phenotype? From the description of the growth conditions (lines 92-95), it appears that it is of the smooth (planktonic) variety. However, clinical isolates are always of the rough type and periodontitis is known to be caused by the rough type.  If the authors have used the smooth type, please provide a reference for periodontitis being caused by smooth Aa.

Line 92-94: “bacterial strains of IC were inoculated ….. daily, for 4 weeks.” The meaning of this is not clear. Inoculated from what? Is it a broth to broth or plate to broth transfer? Microbiologists usually try to avoid broth to broth transfer because that can lead to enrichment of mutants or contamination by other bacteria. 

Line 95: “, for oral contamination” I think it will be better to start a new sentence here.

Line 116: “fresh 24-h inoculum, consisting of the three bacteria” Somewhere in the manuscript, preferably earlier, please specify how the inoculum was made.  I guess, the strains were separately grown, their cell densities were determined and were then mixed to contain a certain cell density of each.

Line 117: Change “but also” to “and also”

Line 118, Figure 1: Purpose of showing pictures of the microfuge tubes is not clear. It does not give any information and actually leads to confusion. There nine tubes shown but only three panels are shown for each time point. Once again, as mentioned above, the procedure for making the inoculum should be clearly specified.

Line 129: “Analyzes” I am not familiar with this word being used as a noun. I think it should be “Analyses”

Line 159-163: It is not clear why a discussion of the result is a part of the figure legend.

Figure 3: What does Day zero mean? Before contamination or immediately after? Why are the neutrophil counts so much different in the three panels for day zero?

Line 160: “The most relevant decrease of neutrophils was observed in group 109” This statement is confusing. If we consider percent decrease, it is the same in 108 and 109 groups but is much greater in the 1010 group.

Line 162: “108” the 8 should be in superscript.

Figure 3 and Figure 4: Since the abbreviations NEU and LYM have not been mentioned anywhere, I think they can be mentioned in the figure legends.  Also, for Figure 4, the unit in the Y-axis should be in parentheses: (K/ml)

Figure 4: Why not have the same Y-axis scale for the three panels so that they can be compared more easily?

Line 197: “Gram-specific” Significance of the word “specific” is not clear.

Line 266: “enzymes tissue destruction” Do you mean “enzymes for tissue destruction”

Line 271: “strains were tested at a concentration of 109.” This is a very high cell density. I understand that you are trying to develop a model for development of PD, but can you also comment on the “real world” situation. Is there any reference for development of PD at a much lower concentration of bacteria for an extended period of time?

Line 275: “acute phase (in group 109) and a chronic phase (in the case of group 1010).” It appears to be counter intuitive that a lower cell concentration causes acute infection with greater effect than a higher cell concentration. Authors, please comment on this. In the reference cited [34] (which I have not read), did the authors observe a similar correlation between inoculum cell density and acute or chronic PD?

Minor editing needed.

Author Response

Please see the attachament.

Thank you!

Reviewer 2 Report

Authors established a rat model of periodontal disease by applying ligatures with the gingival sac contaminated with fresh cultures of Aggregatibacter actinomycetemcomitans, Fusobacterium nucleatum and Streptococcus oralis at concentrations of 10E8, 10E9 and 10E10CFU/ml respectively. The study is of interest. Authors may wish to revise and enhance the manuscript by addressing some specific comments:

1. Use “PD” to represent “periodontal disease” consistently in the entire manuscript after it is first defined at line 40.   

2. A sentence should be provided to justify the selection of F.n and S.o (line 92) as the bacteria strains in the current study, since P.g and T.d. are the triggering agents for PD in adults as stated at Line 63. Also the statement at line 63 is not coherent with the statement at lines 254-255: “we chose the ligation model completed by oral contamination with 3 of the most representative bacteria for periodontitis”. Authors need to make statements consistently!

3. Current dose description was misleading: since the bacteria were used 0.6 mL (line 97) at the concentrations of 108, 109 and 1010/mL, the actual doses should be 6 x 107, 6 x 108 and 6 x 109 respectively. Corrections should be made accordingly. If this reviewer did not interpret the dose correctly, then authors need to make a better description not to misled readers regarding the bacteria doses.

4. Figure 3 showed that NEU values differed drastically among the three dose groups at day 0. Authors should have a sentence or two in the discussion to make some explanation/interpretation of that. For Figure 3, the Y axis for all three panels should be on the same scale for readers to have easy vision comparisons.

5.  Y axis scale should also be the same for the three panels in Figure 5.

6. Figure 6 should present comparable histological images from the three dose groups (as in the middle group). Arrows should be added to each image pointing to the most significant change (neutrophil infiltration).

7. Results of microbiological examination were discussed at lines 191-201  but no data was shown. This is important data that authors should present. Both qualitative (images of slides showing different bacterial staining) and quantitative (number of bacterial colonies  in a specific view area) should be presented. This data would be important to support the overall conclusion that bacterial doses of 109 and 1010/mL, but not 108/mL, induced obvious PD.   

English editing of the entire manuscript is recommended.

Author Response

Thank you!

Round 2

Reviewer 2 Report

Authors made good efforts to revise the manuscript addressing review comments. Revised manuscript is improved. Notably the order of the last two authors was switched in the revised manuscript. Proper confirmation of consent from both authors should be provided.